# Investigation of the Genetic Determinants of Telangiectasia and Solid Organ Arteriovenous Malformation Formation in Hereditary Hemorrhagic Telangiectasia (HHT)

**DOI:** 10.3390/ijms25147682

**Published:** 2024-07-12

**Authors:** Kevin J. Whitehead, Doruk Toydemir, Whitney Wooderchak-Donahue, Gretchen M. Oakley, Bryan McRae, Angelica Putnam, Jamie McDonald, Pinar Bayrak-Toydemir

**Affiliations:** 1Division of Cardiovascular Medicine, Department of Medicine, University of Utah, Salt Lake City, UT 84112, USA; kevin.whitehead@hsc.utah.edu; 2HHT Center, Department of Radiology, University of Utah, Salt Lake City, UT 84112, USA; 3School of Liberal Arts, Tulane University, New Orleans, LA 70118, USA; dtoydemir@tulane.edu; 4Department of Pathology, University of Utah, Salt Lake City, UT 84112, USAangelica.putnam@imail2.org (A.P.); jamie.mcdonald@hsc.utah.edu (J.M.); 5Department of Otolaryngology-Head and Neck Surgery, University of Utah, Salt Lake City, UT 84112, USA; gretchen.oakley@hsc.utah.edu (G.M.O.); bryan.mcrae@hsc.utah.edu (B.M.); 6ARUP Laboratories|Institute for Clinical and Experimental Pathology, 500 Chipeta Way, Salt Lake City, UT 84103, USA

**Keywords:** hereditary hemorrhagic telangiectasia, somatic, biallelic, tissue, telangiectasia, AVM

## Abstract

Telangiectases and arteriovenous malformations (AVMs) are the characteristic lesions of Hereditary Hemorrhagic Telangiectasia (HHT). Somatic second-hit loss-of-function variations in the HHT causative genes, *ENG* and *ACVRL1*, have been described in dermal telangiectasias. It is unclear if somatic second-hit mutations also cause the formation of AVMs and nasal telangiectasias in HHT. To investigate the genetic mechanism of AVM formation in HHT, we evaluated multiple affected tissues from fourteen individuals. DNA was extracted from fresh/frozen tissue of 15 nasal telangiectasia, 4 dermal telangiectasia, and 9 normal control tissue biopsies, from nine unrelated individuals with HHT. DNA from six formalin-fixed paraffin-embedded (FFPE) AVM tissues (brain, lung, liver, and gallbladder) from five individuals was evaluated. A 736 vascular malformation and cancer gene next-generation sequencing (NGS) panel was used to evaluate these tissues down to 1% somatic mosaicism. Somatic second-hit mutations were identified in three in four AVM biopsies (75%) or half of the FFPE (50%) samples, including the loss of heterozygosity in *ENG* in one brain AVM sample, in which the germline mutation occurred in a different allele than a nearby somatic mutation (both are loss-of-function mutations). Eight of nine (88.9%) patients in whom telangiectasia tissues were evaluated had a somatic mutation ranging from 0.68 to 1.96% in the same gene with the germline mutation. Six of fifteen (40%) nasal and two of four (50%) dermal telangiectasia had a detectable somatic second hit. Additional low-level somatic mutations in other genes were identified in several telangiectasias. This is the first report that nasal telangiectasias and solid organ AVMs in HHT are caused by very-low-level somatic biallelic second-hit mutations.

## 1. Introduction

Hereditary Hemorrhagic Telangiectasia (HHT) is an autosomal dominant inherited vascular malformation disorder that occurs in 1 in 5000 individuals [1]. HHT is characterized by recurrent epistaxis ranging from mild to severe; telangiectasia of the nasal mucosa, lips, oral cavity, and/or fingertips; arteriovenous malformations (AVMs) in the brain, lungs, liver, and/or gastrointestinal tract; and/or a family history of HHT. Three or more of these “Curaçao Criteria” are considered diagnostic of HHT [2]. But, given the variable clinical expression and age-related penetrance of clinical manifestations, genetic testing is often useful in making or confirming the diagnosis of HHT.

Mutations in several transforming growth factor-beta (TGF-β) signaling pathway genes cause HHT. Endoglin (*ENG*), activin A receptor type II-like 1 (*ACVRL1*/*ALK1*), and *SMAD4* mutations cause HHT1 (OMIM 187300), HHT2 (OMIM 600376), and the combined Juvenile Polyposis/HHT (JP/HHT) syndrome (OMIM 175050), respectively [3,4,5]. *ENG* and *ACVRL1* mutations account for roughly equal percentages of the disorder and are found in 96.1% of cases with HHT when the consensus Curaçao Criteria are strictly applied [6,7,8]. Analysis of *SMAD4* adds an additional detection rate of 1.3% for a total of 97.4% [9]. Mutations in these genes lead to an underproduction of functional proteins, resulting in excessive abnormal angiogenesis [10].

Patients with HHT harbor these heterozygous germline mutations in every cell of the body, yet vascular malformations occur in only a small minority of vessels, with predilection for specific anatomic locations. The severity of epistaxis and number and location of telangiectases and AVMs vary significantly, even among affected individuals from the same family who have the same genetic mutation [7]. This heterogeneity suggests that additional factors beyond the inherited germline mutation are required to induce vascular malformations. Somatic second-hit mutations play an important role in a number of diseases including vascular malformations [11,12,13]. A single previous report has identified somatic second-hit mutations in dermal telangiectasias in HHT [14]. The mutations reported previously were all found to be in the same gene as the germline heterozygous mutation. It is not known if this mechanism is prevalent, or whether it is also present in the telangiectasia of other clinical locations. It is assumed that such mutations would lead to a loss of heterozygosity and loss of protein expression, which would require the mutation to affect the other allele. Unlike other vascular malformation syndromes [12,13], such a biallelic mechanism has not been proven for the mutations previously observed in HHT. Other reports have identified an important role for somatic mutations that regulate biologic processes such as cell growth, differentiation or angiogenesis that could modify telangiectasia growth or stability [15,16]. A high prevalence of *PIK3CA*-activating mutations has been identified in the vascular malformations seen in both sporadic and familial forms of cerebral cavernous malformation [17].

Furthermore, AVMs in certain locations, in particular the brain, are typically developmental and likely to be present at birth, whereas mucocutaneous telangiectasia are unlikely to be present at birth and tend to develop over time [18]. The age-dependent nature of telangiectasia fits well with the gradual acquisition of somatic mutations over time. Whether somatic mutations contribute to developmental lesions in HHT such as brain [19] or lung AVM [20] that are more likely to be present at birth or early in life and show more stability over time has not been shown.

In this study, we characterized the somatic mutations identified from vascular malformations removed from patients with HHT including nasal and dermal telangiectasias as well as solid organ AVMs. Our technique included a large panel of genes including those known to cause HHT as well as over 700 additional vascular malformation genes with deep sequencing to allow us to detect mosaicism down to at least 1%.

## 2. Results

Tissue specimens from 14 unrelated individuals genetically diagnosed with HHT were evaluated using a custom 736 gene NGS panel to identify the presence of a somatic second hit. The analysis was performed in three stages. First, we performed manual analysis and variant calling of the sequencing data for the genes known to cause HHT (*ACVRL1*, *ENG*, *GDF2*, *SMAD4*) or vascular malformation syndromes with similar phenotypes (*EPHB4*, *RASA1*), as well as *PIK3CA*, which has been shown to modify vascular malformation phenotypes [17]. All variants with a quality score (QC) ≥ to 100 were analyzed using Integrated Genome View (IGV). Second, we used an internal variant caller algorithm (see Section 4). In this analysis, we identified pathogenic or likely pathogenic mutations from a subset of 70 genes (see Section 4) chosen because of a relationship to vascular malformations or their involvement in similar genetic pathways at a level of somatic mosaicism down to 1%. Third, we analyzed all 736 genes (Appendix A), mostly concentrating on damaging hot spot mutations or known pathogenic mutations.

Patient demographics, biopsied tissue details as well as germline and somatic variants are listed in Table 1 (telangiectasias) and Table 2 (AVMs). Thus, 3 of the 14 cases carry *ENG*, and 11 of the cases carry *ACVRL1* germline variants. Further, 15 nasal telangiectasia, 4 dermal telangiectasia, and 9 control (7 nasal and 2 dermal) biopsies from 9 cases were obtained. FFPE tissues from five individuals who had undergone surgical procedures were also obtained and evaluated. Of five cases, three cases had brain arteriovenous malformation resection, one case had lung, and one case had liver and gall bladder tissue samples.

### 2.1. Somatic Mutations Detected in Nasal and Dermal Telangiectasias

We studied a total of 28 biopsies from nine patients that included 15 nasal telangiectasias, 7 nasal control biopsies, 4 dermal telangiectasias, and 2 control dermal biopsies. A heterozygous germline variant was identified in *ENG* or *ACVRL1* in all tissue specimens (Table 1). One individual (Case #9) had only a dermal procedure with three biopsies (two telangiectasia biopsies and one from healthy skin). Two individuals (case 7 and 8) had both nasal and dermal procedures, each with five biopsy and four biopsy samples available (two from nasal telangiectasia, one dermal telangiectasia, and one from each control region, except one dermal control sample for case 7). The other six cases had only nasal procedures.

Pathogenic or likely pathogenic variants were identified in seven of the nine individuals (77.7%, 7 out of 19 telangiectasia samples, 36.8%), always in the same gene that had the germline mutation (Table 1). We identified a pathogenic somatic second hit from at least one telangiectasia sample from every individual with a germline mutation in *ACVRL1*, and none of the individuals with a germline mutation in *ENG*. Three of the seven somatic variants detected resulted in the formation of a stop codon, leading to complete loss of function in *ACVRL1.* Four other variants are missense changes in *ACVRL1*, classified as likely pathogenic according to the accepted guidelines [21].

Somatic mosaicism ranged from 0.68 to 1.96%. Each low-level somatic second-hit variant detected had high-quality variant base pair reads present in both directions, with an average total read depth of 1586X. One case (case 1) had the somatic synonymous *ENG* variant (c.24G>T; p.Leu8Leu). This variant creates a strong acceptor site in the exon with prediction of a splicing defect. However, based on ACMG Classification Guidelines, it is classified as likely benign. All other telangiectasia samples and control biopsies tested negative for a detectable somatic mutation in the seven manually adjudicated genes. None of the samples had a pathogenic or likely pathogenic mutation in the focused subset of 70 additional genes or the broader panel of 736 genes.

### 2.2. Somatic Mutations Identified in AVMs

We obtained tissue samples from five individuals that had surgical resection of AVMs or other tissues. We analyzed three resected brain AVM samples, one lung AVM sample, and a sample of resected liver with focal nodular hyperplasia and adjacent gall bladder. Some of the samples were sequenced twice, for a total of 10 independent sequencing results that were evaluated for somatic variants. A heterozygous germline variant was identified in *ENG* or *ACVRL1* in all FFPE tissue specimens (Table 2). We identified somatic mutations in three of the six tissues (50%), including one clearly pathogenic mutation in the same gene as the germline mutation, one VUS in the same gene as the germline mutation (confirmed on two separate sequencing reactions), and one likely benign synonymous variant in the same gene as the germline mutation (Table 2). A liver with focal nodular hyperplasia and adjacent gall bladder was studied for case 5. The lesion type is probably different from the other four AVM cases. Somatic mutations specifically for AVM tissues were found in three of four cases (75%) by excluding case 5.

Case 1 had removal of an asymptomatic cerebellar AVM. The sample had a pathogenic germline variant in *ENG*. The brain AVM tissue showed 1.96% frequency of a pathogenic variant (c.584_585del, p.Glu195Valfs*138) in the *ENG* gene. This deletion creates a premature stop codon, leading to a predicted loss of function of the protein. Since both germline and somatic second-hit pathogenic variants were in close proximity on the sequence, we could identify the chromosomal positions of these mutations. All 74 reads with the somatic mutation were located on the opposite allele from the germline mutation (Figure 1).

Case 4 had extensive right lower lobe lung AVM with chronic hypoxia and clubbing and had a partial lobectomy. This sample had a pathogenic germline variant in *ACVRL1* c.914C>T, p.Ser305Phe. We obtained duplicate sequencing results from the same block for this sample. Both sequencing reactions showed the same missense variant, c.1220A>T, p.Glu407Val, in the *ACVRL1* gene, with 1.17% and 1.82% frequency. Based on ACMG guidelines, this variant has been classified as VUS. However, prediction programs show that this rare variant may affect the splicing efficiency of the exon. Further investigation would be required to reclassify the variant as pathogenic.

One brain AVM (case 2) with a germline *ENG* c.816+2T>C mutation had a synonymous somatic mutation in *ENG* c.507C>T, p.Leu169Leu that is classified as likely benign. An additional brain AVM (case 3) with a germline mutation in *ENG* (c.640_644del, p.Gly214Glnfs*118 had no identifiable somatic mutations).

Case 5 had right upper quadrant pain and an abnormal Hepatobiliary Iminodiacetic Acid (HIDA) scan, without evidence of cholelithiasis but with an adjacent focal nodular hyperplasia lesion in the liver. The gallbladder and adjacent liver segment were resected laparoscopically. A germline *ACVRL1* c.998G>T, p.Ser333Ile mutation was present in both tissues. No variant was identified in two sequencing reactions from the liver sample, with no somatic mutations identified in the gallbladder tissue.

All AVM tissues tested negative for any other detectable somatic mutations in the seven manually adjudicated genes. None of the samples had a pathogenic or likely pathogenic mutation in the focused subset of 70 additional genes, or the broader panel of 736 genes. All samples were analyzed specifically for *PIK3CA* variants, and no variant was identified in any of the samples.

## 3. Discussion

A previous report identified somatic mutations in dermal telangiectasias from patients with HHT [14]. In this study, we confirm that somatic mutations can be identified from dermal telangiectasia in patients with HHT and, for the first time, document somatic mutations from nasal telangiectasias and solid organ AVMs in patients with HHT. We also confirm that somatic second-hit mutations occur in a biallelic manner.

The mechanism behind AVM and telangiectasia formation is not completely understood, and it is possible that divergent mechanisms contribute to mucocutaneous telangiectasia, as opposed to brain or lung AVM. Telangiectasias are generally not present at birth and form over time, with incidence increasing with age (a pattern reflected by dermal telangiectasia, epistaxis, and liver telangiectasia, leading to high-output heart failure) [22,23]. This disease pattern is consistent with the acquisition of somatic mutations over time that could account for the increasing incidence. In contrast, brain AVM is generally present at birth, and reports of de novo brain AVM formation after birth are extremely rare [19]. This striking difference in the natural history of these HHT-related vascular malformations has led to the hypothesis that AVM and telangiectasia have different underlying mechanisms, calling into question the need for somatic second-hit mutations in the etiology of brain AVM. Here, for the first time, we identify a pathogenic somatic second-hit mutation from a brain AVM of a patient with HHT, suggesting that the underlying mechanism for both brain AVM occurring during development and telangiectasia that develop with aging is shared, and that the loss of heterozygosity for the HHT genes is a common and important disease mechanism.

The second-hit mutations identified in this and previous reports [14] are all in the same gene as the germline-inherited mutation. It has been assumed that these somatic mutations are biallelic—derived from the opposite allele to the germline allele—but current techniques have not been able to confirm this hypothesis. For the first time, we document the biallelic nature of the somatic mutation. We identified a mutation in *ENG* that occurred in close sequence proximity to the germline allele in a patient with brain AVM. Sequence data were able to clearly show that the somatic and germline mutations are on separate alleles (Figure 1). The biallelic loss of *ENG* in this case would lead to a loss of protein expression, and we hypothesize that a loss of heterozygosity in endothelial cells is a central mechanism, leading to vascular malformations in HHT. These somatic mutations are likely random and relatively common, and in order to result in a vascular malformation, they likely occur in the correct (endothelial) cell that is in a vulnerable state. The temporal and tissue specificity of vascular malformations in HHT likely reflects the factors that lead to vulnerable endothelial cells, such as active angiogenesis or inflammation that call upon the BMP9-ACVRL1/ENG-SMAD4 pathway to regulate vascular patterning. The apparent developmental nature of brain AVM is likely to be explained by the temporal specificity of these activating factors, rather than an alternate mechanism that does not require somatic mutations, leading to a loss of HHT gene protein.

Others have reported a high frequency of somatic activating mutations in *KRAS* and *BRAF* in sporadic brain AVM [24] and activating mutations in *MAP2K1* in extracranial AVM [25,26]. We evaluated each brain AVM sample for mutations in *KRAS*, *BRAF*, and *MAP2K1* as well as over 700 additional candidate genes. We looked specifically for the previously described *BRAF* V600E and Q636X mutations, and the *KRAS* G12D and G12V mutations [27] in our samples. We did not identify any activating *KRAS*, *BRAF*, or *MAP2K1* mutations or any other somatic mutations apart from a loss of heterozygosity for the same HHT gene as the germline allele. Our data suggest a divergent mechanism for brain AVM formation in patients with HHT in comparison to sporadic brain AVM. There is evidence that brain AVM in HHT is clinically different than sporadic brain AVM, with greater likelihood of multiple lesions [28], diverse vascular malformation phenotypes in HHT [29], and a different risk of hemorrhage [30]. The genotype differences observed in our series of brain AVM samples in comparison to studies of somatic mutations in sporadic AVM are likely to be an important determinant of the phenotype differences. This suggests that the pathways involved in brain AVM formation and the potential therapeutic approaches are likely different.

This is also the first report of somatic second-hit mutations in nasal telangiectasias. Previous reports have used dermal telangiectasias from patients [14]. Such biopsies are obtained for research purposes, as there is little clinical indication to remove dermal telangiectasia. In contrast, nasal telangiectasias are highly likely to be symptomatic and have long been the subject of a number of ablative therapies to improve symptoms [31]. We have been able to develop a research protocol that can be deployed by our providers during nasal ablative surgery to first obtain research biopsies from the affected mucosa before completing thorough telangiectasia destruction for clinical benefit. This report documents a significant success rate in identifying somatic second-hit mutations from these samples and points to the untapped potential of research using nasal telangiectasia, the single most symptomatic and unwanted lesion encountered by patients with HHT [32].

The presence of somatic second hits in these lesions may explain the extreme variability in the HHT phenotype among individuals in the same family. The observed age-related development of telangiectasia and worsening of nosebleeds related to nasal telangiectases over the lifespan could be influenced by a tertiary mechanism of inflammation or repeated wounding that can lead to the overuse of genetic repair mechanisms, which, over time, could lead to the development of the somatic second hit (similar to cancer) [33,34]. Telangiectasia and AVMs likely only develop under certain specific conditions and are predisposed to occur in certain body locations in those with HHT. For example, case 9 developed 100s of telangiectasias on his arm after undergoing radiation therapy for cancer. This suggests that a somatic mutation may form a vascular lesion if it (1) occurs in an endothelial cell within a specific surrounding cellular environment that is (2) driven in response to an environmental (telangiectasia) or developmental (AVM) trigger.

In the HHT patients in whom a somatic second hit was not found in a telangiectasia or AVM using our NGS method, it is possible that they had a somatic mutation that was not detected because the mosaicism was less than 1%. Snellings et al. identified several telangiectasias with somatic mutations <1%, with the lowest being a 0.46% mutation [14]. Our use of the higher cutoff of 1% somatic mutation and our ability to detect the mutant base in multiple reads in both directions of the NGS data (Figure 1) ensured the confidence that the somatic variants were real. In our analysis, high specificity with the lowest detection limit is 1%. Alternatively, we could be missing certain types of variations that our current NGS exon-specific panel would miss, such as mutations due to a loss of heterozygosity (LOH), deep noncoding splicing aberrations, or promotor variants that would lead to a loss of expression. LOH, a common mechanism of tumor formation in cancer, can occur from large deletions, large chromosomal abnormalities, or mitotic recombination events, all of which cannot be detected using our NGS exon capture technique. It is also possible that non-genetic mechanisms such as a loss of expression due to epigenetic silencing contribute to biallelic loss of function in these lesions.

## 4. Materials and Methods

### 4.1. Subjects

This study was approved by the University of Utah (IRB # 00020480). Subjects were individuals seen at the University of Utah HHT Center of Excellence who met Curacao clinical diagnostic criteria for HHT, had a pathogenic variant in *ENG* or *ACVRL1*, and had a clinical indication for removal or treatment of telangiectasis and/or AVMs. Biopsies of nasal telangiectasia, as well as nearby normal nasal mucosa as control samples from same individual (approximately 3 mm), were collected from 9 unrelated individuals and frozen immediately. These patients were having ENT operations under anesthesia due to epistaxis. A total of 28 biopsies were evaluated. Of them, 15 were from nasal telangiectasia, 4 from dermal telangiectasia, 7 from nasal control region, 2 dermal control biopsies. Age, sex, and HHT clinical findings for each patient are listed in Table 1.

FFPE brain, lung, gall bladder, and liver AVM tissues resected during surgical procedures were obtained on 5 additional patients (brain samples from 3 patients, lung from 1 patient and liver and gall bladder samples from 1 patient). Three of the five patients had *ENG*; two of them had *ACVRL1* germline mutation (Table 2).

### 4.2. Next-Generation Sequencing and Data Analysis

Genomic DNA was extracted from fresh/frozen tissue using a Puregene kit (Qiagen, Valencia, CA, USA). Genomic DNA (1 µg) was sheared to 300 bp fragments on a Covaris LE220 sonicator (Covaris, Inc., Woburn, MA, USA), and custom adapters for the Illumina platform were added using the KAPA Hyper Prep Kit (Roche, Branchburg, NJ, USA). Genomic libraries were enriched for exons and exon/intron boundaries of 736 genes implicated in cancer and vascular malformations (including *ENG*, *ACVRL1*, *SMAD4*, *GDF2*, *RASA1* and *EPHB4*) using xGen Lockdown probes and the xGen Hybridization and Wash kit (Integrated DNA Technologies, Inc., Coralville, IA, USA). The 736 genes included in the panel are listed in Appendix A. Final libraries were sequenced on the NovaSeq 6000 instrument with 2 × 150 base paired end reads (Illumina, San Diego, CA, USA).

Sequences were aligned to the human genome reference (hg19) sequence using the Burrows–Wheeler Alignment tool (BWA mem 0.7.17) with default parameters [35]. PCR duplicates were removed based on positional information and a 6 bp UMI sequence in the adapter using an in-house UMI Aware Mark Duplicates tool from the Genome Analysis Tool Kit (GATK) [36]. The Duplicate Removed Bam was realigned using GATK Indel Realigner [36]. Variants were then called by a combination of LOFREQ [37] (SNVs only), and SCALPEL [38] (small Indels), MANTA [39] (large Indels) and PINDEL [40] for FLT3-specifc insertions/duplications/deletions. Variants were annotated for gene position and various external database information (such as dbSNP, Gnomad, Cosmic) using software from GenomOncology version 23Q3. Variants were stored in a MongoDB database and visualized using an in-house User Interface called NGS.Web. To be able to detect 1% sensitivity, samples were deeply sequenced. Coverage of NGS reads was around 1500 reads (900 reads, 3688).

## 5. Conclusions

We confirm that somatic mutations are prevalent in the vascular malformations of HHT including nasal and dermal telangiectasia as well as AVMs. Our data suggest that the biallelic loss of *ENG* and *ACVRL1* is required for the development of both congenital AVMs and acquired telangiectasia vascular malformation lesions observed in HHT. This is the first report of somatic mutations driving lesion formation in AVMs and nasal telangiectasia in HHT patients.

## Figures and Tables

**Figure 1 ijms-25-07682-f001:**
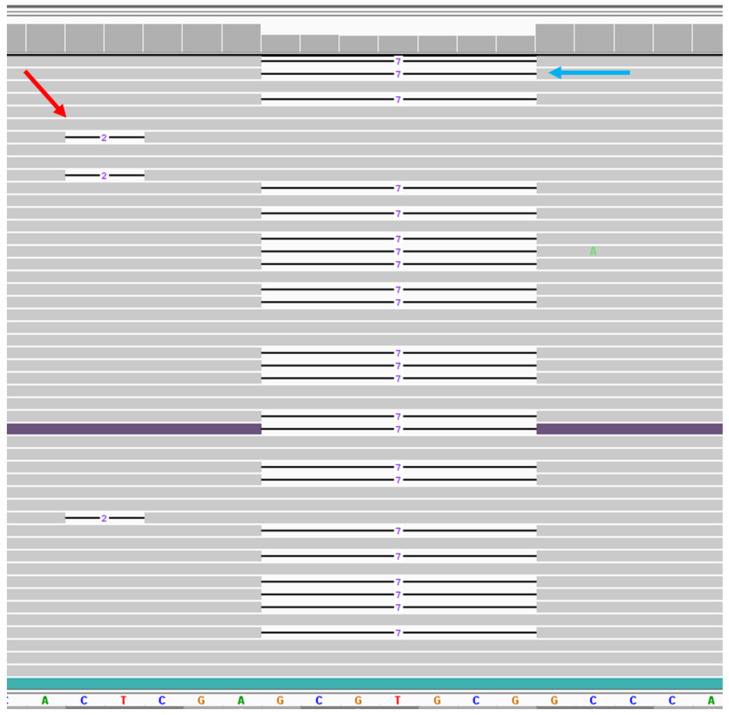
NGS data of the somatic second-hit variant in telangiectasia tissue. Viewing the *ENG* sequence region by Integrated Genomics View (IGV) for case 1. Bottom panel shows reference sequences of the region where both variants were located. Gray bar shows the reads match with reference sequences. Line indicates the deletion. On the left, red arrow shows the representation of 3 reads out of 74 reads for the 2 bp somatic deletion. On the right, blue arrow shows the representation of 23 reads out of 1415 reads for 7 bp germline deletion. Both deletions are not in phase with each other.

**Table 1 ijms-25-07682-t001:** Clinical and molecular findings of fresh nasal and dermal tissue samples from HHT cohort.

Case #	Age, Sex	Phenotype	Germline Mutation	Biopsies	Tissue Details	Somatic Mutation	Somatic VAF (%) NGS Reads	Classification
1	32 M	E, T, F	*ENG* c.1146C>A, p.Cys382*	4 (3NT, 1NC)	Nasal T (Left septum)	*ENG* c.24G>T, p.Leu8Leu	1.03% (10/974)	LB
					Nasal T (Right wall)	None detected	-	
					Nasal T (Left septum)	None detected	-	
					Nasal C (Left inferior turbinate)	None detected	-	
2	50 M	E, T, F, L, H	*ENG* c.1687G>T, p.Glu563*; c.1687-7C > T	2 (1NT, 1NC)	Nasal T (Right lateral nasal floor)	None detected	-	
					Nasal C (Right inferior turbinate)	None detected	-	
3	30 M	E, T, F	*ACVRL1* c.430C>T, p.Arg144*	2 (1NT, 1NC)	Nasal T (Right hard palate)	*ACVRL1* c.1411T>C, p.Cys471Arg	1.39% (18/1293)	LP
					Nasal C (Left hard palate)	None detected	-	
4	68 M	E, T, F, H	*ACVRL1* exon 10 deletion	1 (1NT)	Nasal T (Right nasal valve)	*ACVRL1* c.598C>G, p.Arg200Gly	0.96% (11/1147)	LP
5	45 F	E, T, F, H, PAH	*ACVRL1* c.998G>T, p.Ser333Ile	3 (2NT, 1NC)	Nasal T (Right septum)	*ACVRL1* c.1461G>C, p.Lys487Asn	1.46% (17/1167)	LP
					Nasal T (Left lateral wall)	None detected	-	
					Nasal C (Right septum)	None detected	-	
6	43 M	E, T, F	*ACVRL1* c.1361_1375del, p.Arg454_Asp458del	4 (3NT, 1NC)	Nasal T (Left inferior turbinate)	*ACVRL1* c.611T>A, p.Leu204*	1.37% (26/1893)	P
					Nasal T (Right inferior turbinate)	None detected	-	
					Nasal T (Right inferior turbinate)	None detected	-	
					Nasal C (Left inferior turbinate)	None detected	-	
7	66 M	E, T, F, Possible H	*ACVRL1* c.472_473del, p.Gly158Argfs*10	4 (2NT, 1DT, 1NC)	Nasal T (Right middle turbinate)	None detected	-	
					Dermal T (Right forehead)	*ACVRL1* c.988G>A (p.Asp330Asn)	0.68% (12/1758)	LP
					Nasal T (Right inferior turbinate)	None detected	-	
					Nasal C (left inferior turbinate)	None detected	-	
8	57 M	E, T, F	*ACVRL1* c.1232G>A, p.Arg411Gln	5 (2NT, 1DT, 1NC, 1DC)	Nasal T (Right inferior turbinate)	None detected	-	
					Dermal T (Right index finger) HHT21T4	None detected	-	
					Nasal T (Left anterior septum)	*ACVRL1* c.274_275del; p.(Ser92Profs*76)	0.84% (16/1900)	P
					Nasal C (Right inferior turbinate)	None detected	-	
					Dermal C (Right index finger)	None detected	-	
9	39 M	E, T, F, H	*ACVRL1* c.1232G > A, p.Arg411Gln	3 (2DT,1DC)	Dermal T (developed after radiation therapy in treated region)	*ACVRL1* c.429_430delinsTT, p.Arg144*	1.96% (38/1941)	P
					Dermal T	None detected	-	
					Dermal C	None detected	-	

C (Control); DC (Dermal biopsy from control region); DT (Dermal Telangiectasia); E (Epistaxis); F (Family History); H (Hepatic AVM); L (Lung AVM); LB (Likely Benign); LP (Likely Pathogenic); NC (Nasal biopsy from control region); NT (Nasal Telangiectasia); PAH (Pulmonary arterial hypertension); T (Telangiectasia); VAF (Variant Allele Frequency); P (Pathogenic). Reference sequences NM_000020.2 and NM_001114753.1 were used to annotate *ACVRL1* and *ENG*, respectively.

**Table 2 ijms-25-07682-t002:** Clinical and molecular findings of FFPE samples from HHT cohort.

#	Germline Mutation	Tissue	Sequencing Data Point	Somatic Tissue	Classification	Somatic VAF (%) NGS Reads
1 *	*ENG* c.574_580del; p.Arg192Serfs*28	Brain	One block One seq	*ENG*; c.584_585del; p.Glu195Valfs*138	P	1.95% 74/3783
2	*ENG* c.816+2T>C	Brain	One block One seq	*ENG*; c.507C>T; p.Leu169=	LB	0.84% 10/1185
3	*ENG* c.640_644del; p.(Gly214Glnfs*118)	Brain	One block Duplicate seq	None	-	-
4	*ACVRL1* c.914C>T; p.Ser305Phe	Lung	One block Duplicate seq	*ACVRL1*; c.1220A>T; p.Glu407Val **	VUS	1.17% 43/3688 and 1.82% 44/2422
5	*ACVRL1* c.998G>T; p.Ser333Ile	Liver	Two blocks	None	-	-
		Gall bladder	Two blocks	None	-	-

*: Somatic second-hit is identified on the opposite allele of the germline mutation. ** Although this variant is missense change, based on prediction programs this variant has possible splice defect. LB: Likely Benign. VUS: Variant of Uncertain Significance. P: Pathogenic.

## Data Availability

The raw data supporting the conclusions of this article will be made available by the authors on request.

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
