# Peer review of "Investigation of the Genetic Determinants of Telangiectasia and Solid Organ Arteriovenous Malformation Formation in Hereditary Hemorrhagic Telangiectasia (HHT)"

_ijms, 2024, doi:10.3390/ijms25147682_

Round 1

Reviewer 1 Report

Comments and Suggestions for Authors

Author Response

Comments concerning the subject

Whitehead et al present a case series of HHT patients in whom a potential second-hit somatic mutation is searched. This is the first report of second-hit mutations in nasal telangiectasia and solid organ arteriovenous malformations in the pathogenesis of HHT, that is worth to publish.

The introduction, the study design and the discussion are each well interpreted.

I have only one comment regarding the content: why did the authors interchange the usual order of a manuscript’s sections? The order of introduction – results – discussion – materials and methods is rather unusual. I propose to restore the conventional order or introduction – materials and methods - results – discussion. I am happy to change the order. However, it is in the journal’s instructions specifically put the Materials and Methods at the end. I may leave this decision to editors.

Formal comments

  1. Please use the forms „second-hit” (recommended) or „second hit”, „teleangiectases” (recommended) or „teleangiectasias” consistently. I have changed to second-hit throughout to text. I will leave the decision for telangiectasia word to editors. I believe both should be correct.
  2. Line 94: „Integrated Genome Analyzer (IGV)”. IGV is the abbreviation of Integrated Genomics View, as clarified in the legend of Figure 1. Fixed it thanks
  3. Lines 160-163: „Case #1 had removal of an asymptomatic cerebellar AVM. The sample had a pathogenic germline variant in ENG c.574_580del, p.Arg192Serfs*28. NGS data analysis from the brain AVM tissue showed 1.96% frequency (74 of 3783 reads) of a pathogenic variant (c.584_585del, p.Glu195Valfs*138) in the ENG gene.” This information is redundant as Table 2 contains it already. This sentence should be deleted or significantly shortened. This sentence has been shortened. "The sample had a pathogenic germline variant in ENG. The brain AVM tissue showed 1.96% frequency of a pathogenic variant (c.584_585del, p.Glu195Valfs*138) in the ENG gene".
  4. Line 38: „three patients” instead of „three patientLine 308 has been fixed three patients

Reviewer 2 Report

Comments and Suggestions for Authors

Thank you very much for this very interesting study which corroborates the second hit mutation to explain the appearance of HHT manifestations and provides a good genetic argument in favor of a biallelic mutation of ENG in a cerebral AVM. The discussion on the different temporalities of AVMs and telangiectasias is well conducted.

I only have one comment and one regulatory point.

1) Could you better explain the role attributed to the 1% threshold? The current wording is ambiguous, it can sometimes be interpreted as a lower threshold, sometimes as an upper threshold.

2) Please answer the following questions:

Institutional Review Board statement: (this is given in the text but should be informed here)

Statement of informed consent:  

Data Availability Statement: 

Conflicts of interest:

Author Response

Thank you very much for this very interesting study which corroborates the second hit mutation to explain the appearance of HHT manifestations and provides a good genetic argument in favor of a biallelic mutation of ENG in a cerebral AVM. The discussion on the different temporalities of AVMs and telangiectasias is well conducted.

I only have one comment and one regulatory point.

1) Could you better explain the role attributed to the 1% threshold? The current wording is ambiguous, it can sometimes be interpreted as a lower threshold, sometimes as an upper threshold. Although our analysis can detect 0.5%, we used 1% as a threshold to increase the specificity. For this reason, 1% is not upper or lower threshold. We have added a sentence to line 285for clarity. In the HHT patients in whom a somatic second -hit was not found in a telangiectasia or AVM using our NGS method, it is possible that they had a somatic mutation that was not detected because the mosaicism was less than 1%. Snellings et al. identified several telangiectasia with somatic mutations <1%, with the lowest being at 0.46% mutation [14]. Our use of the higher cutoff of 1% somatic mutation and our ability to detect the mutant base in multiple reads in both directions of the NGS data (Figure 1 and Supplemental Figure 1) ensured the confidence that the somatic variants were real. “In our analysis, high specificity with the lowest detection limit is 1%.

 2) Please answer the following questions” We have updated the manuscript with following info.

Institutional Review Board statement: (this is given in the text but should be informed here): This study was approved by the University of Utah (IRB # 00020480). 

Statement of informed consent:  Informed consent was obtained from all subjects involved in the study

Data Availability Statement: The raw data supporting the conclusions of this article will be made available by the authors on request.

Conflicts of interest: None

Reviewer 3 Report

Comments and Suggestions for Authors

The manuscript entitled Investigation of the Genetic Determinants of Telangiectasia and Solid Organ Arteriovenous Malformation Formation in Hereditary Hemorrhagic Telangiectasia (HHT) is an original article. The authors analyzed multiple affected tissues from fourteen individuals to assess the genetic mechanism of AVM formation in Hereditary Hemorrhagic Telangiectasia (HHT). They concluded that somatic mutations are prevalent in the vascular malformations of HHT, including nasal and dermal telangiectasia as well as AVMs. It seems that for the development of both, congenital AVMs and acquired telangiectasia vascular malformation lesions, bi-allelic loss of ENG and ACVRL1 are required.

The manuscript is well written. This study seems important because for the first time is stated that nasal telangiectasias and solid organ AVMs in HHT are caused by very low level somatic biallelic second hit mutations.

However, this group of researchers included seven of the studies published by them in the list of references (n=40).All seven references are on the same theme, but not all are probably mandatory to be cited in this manuscript (for example 7, 8 or 11). Could you comment this?

Author Response

The manuscript entitled Investigation of the Genetic Determinants of Telangiectasia and Solid Organ Arteriovenous Malformation Formation in Hereditary Hemorrhagic Telangiectasia (HHT) is an original article. The authors analyzed multiple affected tissues from fourteen individuals to assess the genetic mechanism of AVM formation in Hereditary Hemorrhagic Telangiectasia (HHT). They concluded that somatic mutations are prevalent in the vascular malformations of HHT, including nasal and dermal telangiectasia as well as AVMs. It seems that for the development of both, congenital AVMs and acquired telangiectasia vascular malformation lesions, bi-allelic loss of ENG and ACVRL1 are required.

The manuscript is well written. This study seems important because for the first time is stated that nasal telangiectasias and solid organ AVMs in HHT are caused by very low level somatic biallelic second hit mutations.

However, this group of researchers included seven of the studies published by them in the list of references (n=40).All seven references are on the same theme, but not all are probably mandatory to be cited in this manuscript (for example 7, 8 or 11). Could you comment this? Thank you very much for the input. We have changed the references.  However, if reviewers think we need to delete or change more we are happy to do so. Our group has been publishing on HHT for a long time. Now we are under the 20% journal rule for referring to our own work.
